# Low-Dose and Long-Term Olaparib Treatment Sensitizes MDA-MB-231 and SUM1315 Triple-Negative Breast Cancers Spheroids to Fractioned Radiotherapy

**DOI:** 10.3390/jcm9010064

**Published:** 2019-12-26

**Authors:** Clémence Dubois, Fanny Martin, Chervin Hassel, Florian Magnier, Pierre Daumar, Corinne Aubel, Sylvie Guerder, Emmanuelle Mounetou, Frédérique Penault-Lorca, Mahchid Bamdad

**Affiliations:** 1Université Clermont Auvergne, Centre Jean Perrin, INSERM, U1240, Imagerie Moléculaire et Stratégies Théranostiques, F-63000 Clermont Ferrand, France; clemence.dubois@uca.fr (C.D.);; 2Université Clermont Auvergne, Institut Universitaire de Technologie, INSERM, U1240, Imagerie Moléculaire et Stratégies Théranostiques, F-63000 Clermont Ferrand, France; 3Département de Radiothérapie, Centre Jean Perrin, F-63000 Clermont Ferrand, France; 4Centre Hospitalier Universitaire Purpan, Centre de Physiopathologie de Toulouse Purpan, INSERM, UMR 1043/CNRS UMR 5282, Antigen Presenting Cells and CD4 T cell responses, F-31024 Toulouse, France; 5Service de Physique Médicale, Centre Jean Perrin, F-63000 Clermont Ferrand, France; 6Université Clermont Auvergne, Faculté de Médecine, INSERM, U1240, Imagerie Moléculaire et Stratégies Théranostiques, F-63000 Clermont Ferrand, France

**Keywords:** Triple-Negative Breast Cancer, SUM1315, MDA-MB-231, PARPi Olaparib, fractioned radiotherapy, co-treatment, the three-dimensional cell culture, spheroid, transcriptomic analysis

## Abstract

The Triple-Negative Breast Cancer subtype (TNBC) is particularly aggressive and heterogeneous. Thus, Poly-ADP-Ribose Polymerase inhibitors were developed to improve the prognosis of patients and treatment protocols are still being evaluated. In this context, we modelized the efficacy of Olaparib (i.e., 5 and 50 µM), combined with fractioned irradiation (i.e., 5 × 2 Gy) on two aggressive TNBC cell lines MDA-MB-231 (BRCAness) and SUM1315 (BRCA1-mutated). In 2D cell culture and for both models, the clonogenicity drop was 95-fold higher after 5 µM Olaparib and 10 Gy irradiation than Olaparib treatment alone and was only 2-fold higher after 50 µM and 10 Gy. Similar responses were obtained on TNBC tumor-like spheroid models after 10 days of co-treatment. Indeed, the ratio of metabolic activity decrease was of 1.2 for SUM1315 and 3.3 for MDA-MB-231 after 5 µM and 10 Gy and of only 0.9 (both models) after 50 µM and 10 Gy. MDA-MB-231, exhibiting a strong proliferation profile and an overexpression of AURKA, was more sensitive to the co-treatment than SUM1315 cell line, with a stem-cell like phenotype. These results suggest that, with the studied models, the potentiation of Olaparib treatment could be reached with low-dose and long-term exposure combined with fractioned irradiation.

## 1. Introduction

Breast cancer is a complex disease presenting various phenotypes with distinct prognosis and sensitivity to drugs [1]. Among the multiple classifications, the Triple-Negative breast cancers (TNBC) are characterized by the absence of estrogen and progesterone receptor expression, the absence of Human-Epidermal-Receptor Type 2 (HER2) overexpression, a high proliferation, and by the expression of the cytokeratins 5 and 6 [2]. Furthermore, approximately 10–15% of TNBC tumours carry hereditary mutations in the BReast CAncer early onset 1 and 2 genes (BRCA1/2 genes), which are responsible for hereditary predisposed cancers developments [3,4,5]. Given this aggressive phenotype, a great proportion of TNBC tumours recur within 3 years after conventional anticancer treatments such as surgery, chemotherapy or radiotherapy. Indeed, these TNBC tumours temporarily benefit from neo-adjuvant or adjuvant therapies, but then a rapid metastatic relapse occurs in almost 60% of the patients [6]. As a result, many efforts have been made to improve the prognosis of patients. Indeed, the treatment protocols have been optimized and targeted therapeutic agents exploiting the specificities of TNBC tumours developed, such as Poly-ADP-Ribose-Polymerase 1 inhibitors (PARPi such as Olaparib/Lynparza^®^, AstraZeneca) [2,7].

PARPi constitute a class of targeted therapeutics acting on DNA repair systems using the synthetic lethality concept. These open up new prospects for improving the treatment of TNBC aggressive subtype [8]. Since February 2019, the Lynparza^®^ (Olaparib) was adopted by the Committee for Medicinal Products for Human Use (CHMP from European Medicines Agency) for the treatment of adult patients with germline BRCA1/2-mutations, who have HER2-negative locally advanced or metastatic breast cancer. In addition to being used as monotherapy treatment agents, PARPi may also be used in combination with chemotherapy or radiotherapy [8]. These combination methods exhibit a strong rationale due to the inhibition of DNA homologous recombination repair system induced by PARPi and the generation of DNA single and double-strand breaks induced by irradiation and are still being studied in the preclinical setting [9,10].

The pipeline of drug development is composed of several preclinical and clinical phases. The preclinical setting is divided into in vitro and in vivo experiments which goals are the determination of drug efficacy and toxicity. Recently, the three-dimensional cell culture (3D) has been integrated into these preclinical processes after the monolayer in vitro cell culture experiments. The 3D cell culture (spheroid) appears to be an adapted tool for the evaluation of drug efficacy for drug development and optimization in Oncology. Indeed, these 3D models mimic more precisely the tumour microenvironment by the formation of genetical, biochemical and phenotypical gradients, resembling intratumoral gradients and microenvironment [11,12,13]. 

In recent years, in order to optimize the treatment of these aggressive breast cancers, our research group has focused its works on the modeling of the balance between cytotoxicity/resistance of TNBC cell lines, in response to PARPi [14,15]. In this context, these works aimed to determine the effectiveness of a concomitant treatment combining increasing concentrations of Olaparib (PARPi) and fractioned irradiation (5 × 2 Gy) on two TNBC models i.e., MDA-MB-231 (wild-type BRCA1 gene, loss of heterozigoty) [16] and SUM1315 (BRCA1-mutated) [16] cultured in monolayer and 3D cell culture conditions. We first determined the efficacy of this co-treatment by the analysis of (i) DNA double strand breaks induction, (ii) cell survival decrease and (iii) clonogenic survival decrease, in 2D cell cultures. These tests allowed us to further validate this combination on tumor-like models (3D cell cultures), treated at long-term with specific efficient Olaparib concentrations, and daily irradiated (5 × 2 Gy). Finally, an in silico transcriptomic approach highlighted the different cell-line patterns leading to a promising perspective for the treatment of aggressive/metastatic breast cancer tumours.

## 2. Materials and Methods 

### 2.1. Cell Lines and Culture

SUM1315 cell line (Asterand^®^, MO2) was cultured in Ham’s F12 medium supplemented with 5% decomplemented fetal calf serum, 10 mM HEPES buffer, 20 μg/mL gentamycin, 10 ng/mL EGF and 4 μg/mL insulin. MDA-MB-231 cell line (HTB-26^TM^, ATCC^®^) was cultured in RPMI medium 1640 supplemented with 10% decomplemented fetal calf serum and 20 μg/mL of gentamycin. Both cell lines were maintained at 37 °C under 5% CO_2_ in a humidified incubator. The absence of mycoplasma contamination was checked before every experiment by the mycoplasmacheck test (GATC Biotech, Konstanz, Germany). 

According to the experiments, cells were either seeded in Ibitreat microscopy chambers (Foci H2AX assay), six-well flat bottom microplates (clonogenic survival test) or 96-well flat-bottom microplates (cell survival test) at a concentration of 5000 cells/cm^2^ and let in the incubator for 24 h before treatments. 

#### Spheroid Formation

For spheroid formation and treatments, the liquid overlay technique (LOT) was performed in Ultra-Low-Attachment round-bottom microplates according to our previous works [13].

### 2.2. Olaparib Treatment

Olaparib (Carbosynth^®^) was solubilized in Dimethylsulfoxide (DMSO) at a concentration of 100 mM. For all experiments, the solubilized Olaparib was diluted in the culture medium at increasing concentrations with a final and constant DMSO concentration of 0.1%. Parallely, DMSO 0.1% controls without Olaparib were also performed. For Olaparib treatments and co-treatment (Olaparib/irradiation), Olaparib was added once, three hours before the first radiotherapy session. The treatment lasted for 24 h, 72 h or 120 h in 2D experiments, and for 6, 8 or 10 days in 3D experiments. The absence of DMSO 0.1% toxicity was systematically checked prior to results analysis.

#### Irradiation 

The cells were irradiated at a daily dose of 2 Gy for one to five days, on a linear electron accelerator with a 6 MegaVolts X-ray beam used for routine treatment in clinic (Truebeam^®^, Varian Medical Systems, Palo Alto located in Centre Jean Perrin, Clermont-Ferrand, France). The dose rate applied to the accelerator was 400 Monitor Units per minute (MU/min), which corresponds to the dose rate conventionally used in radiotherapy. For this, the cells were placed 3 cm deep inside a 13 cm thick PolyMethyl MethAcrylate (PMMA) phantom in order to obtain a homogeneous distribution of dose and respect the charged particle equilibrium conditions. A computed tomography (CT) scanner examination of this phantom was performed with each device containing the irradiated cells. The dose was prescribed in the middle of the volume containing the cells and the calculation was performed with the Anisotropic Analytical Algorithm (AAA). The irradiated plates remained in the incubator outside of radiotherapy treatment session. The non-irradiated culture plates with or without Olaparib were maintained in the same experimental conditions.

### 2.3. Experiments in Monolayer Cell Culture 

#### 2.3.1. DNA DSBs Induction: Gamma H2AX Immunofluorescent Markings

Cells were treated with 5 and 50 µM Olaparib for 3 h and then were irradiated with a single X-ray dose of 2 Gy. After one hour, cells were fixed in 4% paraformaldehyde, permeabilized in ethanol, incubated with a saturation solution and then with the primary antibody (1/250, Anti-phospho-histone H2AX, Ser 139 clone JBW301) or the isotype control (1/1000, isotype IgG1a mouse). Cells were then incubated with the anti-mouse secondary antibody (1/800, Alexa Fluor 488 gloat anti-mouse) and with Hoechst 33258 for nuclear counterstaining. Finally, cells were dried, mounted and observed with Cytation^TM^3MV plate reader (Biotek, Winooski, VT, USA). The number of γH2AX foci was calculated on >10 cellular fields for each condition with ImageJ^®^ software, representing at least more than hundred cells. The results were represented as the mean number of γH2AX foci per nucleus with their standard deviation (s.d.).

#### 2.3.2. SulfoRhodamine B Survival Test

Cells were treated with increasing concentrations of Olaparib (0.1 to 100 µM) for 24 h, 72 h or 120 h, corresponding to concomitant doses of radiotherapy of 2, 6 and 10 Gy, respectively. After each treatment endpoint, cells were fixed with a trichloroacetic acid (TCA) solution for 1 h, washed in water, dried and incubated with a SRB solution in 1% (v/v) acetic acid for 15 min. Again, cells were washed with 1% acetic acid, dried and a 10 mM Tris-base buffer solution was finally added to the wells for 30 min. The absorbance of SRB dye dissolved in each well was determined at 540 nm with Cytation3MV plate reader (Biotek, Winooski, VT, USA). Cell survival was calculated for each treatment condition by the comparison of 24 h, 72 h and 120 h aged untreated control wells with the following formula:Cell survival (%) = (OD_540 nm_^treated wells^/OD_540 nm_^no X-Ray/no Olaparib = control^) × 100.(1)

The results are presented as mean viability of *n* > 45 wells ± their standard deviation (s.d.). 

#### 2.3.3. Clonogenic Survival Test 

The clonogenic potential of cells following a treatment with irradiation alone or combined with Olaparib (0.5, 5 and 50 µM) was evaluated after cumulative X-ray doses of 2, 6 or 10 Gy, corresponding to concomitant Olaparib exposures of 24 h, 72 h of 120 h. After each treatment endpoint, cells were recovered by trypsinisation, enumerated and re-seeded into new plates at an adapted concentration. The number of colonies formed was determined after nine doubling times (objective 10X). The Plating Efficiency (PE) corresponding to the cell repopulation factor after each treatment condition was calculated as follows: PE = Number of colonies formed/Number of seeded cells at T_0_.(2)

Then, the survival fraction after each X-ray dose was calculated compared to the corresponding untreated control of each dose (= No X-ray and No Olaparib treatment) with the following formula:Survival fraction (%) = PE treated condition ^(X-ray dose ± Olaparib)^/PE control ^(no X-ray/no Olaparib)^ of each corresponding X-ray dose.(3)

The values of the clonogenic survival were expressed as mean survival ± their standard deviation (s.d.) of *n* = 5 replicates.

### 2.4. Experiments in 3D Cell Culture 

#### 2.4.1. Spheroid Treatment

Spheroids aged of 3 days were treated with 5 and 50 µM of Olaparib for 6, 8 and 10 days, corresponding to concomitant X-ray doses of 2 Gy (1 session), 6 Gy (three successive daily sessions) and 10 Gy (five successive daily sessions), respectively. 

#### 2.4.2. Spheroid Growth Monitoring

The size of spheroids after each treatment endpoint (6, 8 and 10 days) was monitored with the Cytation^TM^3MV microplate reader (Biotek, Winooski, VT, USA) using the cellular analysis algorithm of the Gen 5 software (version 03, Biotek, Winooski, VT, USA). Results were expressed as the mean spheroid size of each treatment condition (*n* > 45) with their standard deviation (s.d.). 

#### 2.4.3. Spheroid Metabolic Activity Assessment With the Resazurin Test 

Spheroids from every treatment condition were transferred in a new microplate containing 60 µM resazurin in PBS. The Fluorescence Intensity (FI) corresponding to the amount of resorufin formed after 17 h incubation was quantified in each well with Cytation3^MV^ plate reader (Biotek, Winooski, VT, USA). This allowed the determination of the percentage of spheroid metabolic activity as controls calculated as follows: Metabolic activity (%) = FI treated spheroid ^(X-ray ± Olaparib or Olaparib ± X-ray)^/FI control spheroid ^(no Olaparib/no X-ray)^. (4)

The results were presented as mean spheroid metabolic activity of each treatment condition (*n* > 16 spheroids) ± their standard deviation (s.d.). 

#### 2.4.4. Spheroid Viability and Mortality Fluorescent Profile (Live/Dead)

Spheroids of each treatment condition were harvested, rinsed twice in PBS and incubated with 4 µM ethidium-homodimer (Etdh-1, red fluorescence, dead cells) and 2 µM Calcein-AM (green fluorescence, viable cells) for 45 min. The fluorescence of each fluorophore was then imaged with Cytation3MV plate reader (Biotek, Winooski, VT, USA). For the image analysis, same exposure time, LED intensity and gain were programmed for all image acquisitions.

### 2.5. Transcriptomic Analysis of TNBC Cell Lines

All available MDA-MB-231 and SUM1315 transcriptomic data from different studies were collected from the NCBI public dataset GEO DataSet. Accession number, sample and sequencing information are indicated in the Table 1 below:

Fastq files were analyzed on Genotoul Bio-informatic cluster. Input reads for each sample were aligned to the Human (GRCh38) reference genome using TopHat [20]. Two-dimensional principal component analysis (PCA) of all data was performed to assess quality samples. It depicted that the first principal component represents 63% of variance and separated the two cell lines. The R package Deseq2 was used to analyze the statistically significant Differential Expressed Genes (DEG) between MDA-MB-231 and SUM1315 cell lines. An adjusted *p* value of <0.1 was the cutoff to select specifically DEG between the two cell lines. This DEGs list was subject as query to Ingenuity Pathway Analysis (IPA, QIAGEN Inc., Hilden, Germany). IPA permits to infer functional and cellular processes that are modified between these two cell lines using the Ingenuity Knowledge Base. Two statistical indexes (*p*-value and z-score) are determined for each prediction. The *p*-value is calculated by a Fisher exact test and permits to identify significantly enriched pathways. The IPA z-score is the statistical measure of the match between altered genes and the inferred biological function. z-score let to predict the direction of change of a function. Indeed, a cellular function activity is increased when z-score is >2 and decreased when z-score <−2.

### 2.6. Statistical Analysis

All the data set was statistically compared with a Student’s two-sided test, where significative values are indicated in the graphs as: * *p* < 0.05, ** *p* < 0.01, *** *p* < 0.001, **** *p* < 0.0001 and ***** *p* < 0.00001. The level of interaction between all treatment conditions was determined with a multivariate ANOVA analysis (mixed model), performed with the SEM software (developed by F.K., Centre Jean Perrin, Clermont-Fd, France). 

## 3. Results

### 3.1. Olaparib and Radiotherapy Induce DNA Double Strand Breaks in TNBC Cell Lines 

For MDA-MB-231 cell line (Figure 1A), after a 4 h treatment with Olaparib alone, the number of gH2AX foci per nuclei doubled in presence of 5 µM (8 ± 0.7 than 0.1% DMSO controls 4 ± 0.4, *p* = 10^−^^7^) and tripled after 50 µM (13 ± 1.1, *p* = 10^−12^). After 2 Gy irradiation alone, the number of foci increased up to 19 ± 1.1 (*p* = 10^−26^). In presence of the co-treatment, this parameter kept increasing significantly than irradiation alone or Olaparib treatment alone with 23 ± 1.3 and 32 ± 1.6 foci per nuclei (for 5 µM/2 Gy and 50 µM/2 Gy, respectively) (Figure 1A). Parallely, same results were obtained for SUM1315 cell line (Figure 1B), with a 3.2-fold increase after 50 µM Olaparib treatment (15 ± 1.9 than 0.1% DMSO controls 5 ± 0.5, *p* = 10^−7^), and a 1.8-fold increase after irradiation and Olaparib treatment (31 ± 2.9 for 50 µM than irradiated controls 17 ± 1.1, *p* = 10^−5^)(Figure 1B). These results suggest a cumulative effect of the co-treatment in terms of DNA DSBs induction after 4 h of treatment, probably related to both the irradiation and the Olaparib treatment impact. We then modelized the impact of the co-treatment on 2D cell survival after one to five days of Olaparib exposure and 2 to 10 Gy cumulative X-ray doses. 

### 3.2. The Cytotoxicity of Radiotherapy Is Optimized by Olaparib and is Potentiated in Presence of Low Doses of Olaparib

#### Impact on TNBC Cell Survival in 2D Cell Culture

The cell survival of both MDA-MB-231 and SUM1315 cell lines was analyzed by an SRB survival test, either after treatment with 0.1–100 µM Olaparib concentrations for 24, 72, or 120 h, or combined with 2, 6 or 10 Gy cumulative irradiation doses. The 2 Gy radiation was prescribed daily. All data are presented in Appendix A. In order to determine the impact of each analyzed parameter on cell survival variance, i.e., (i) the type of cell line (MDA-MB-231, SUM1315), (ii) the Olaparib concentration (0.1–100 µM), (iii) the Olaparib time of exposure (24–120 h) and (iv) the irradiation dose (2–10 Gy), the overall cell survival results from both cell line models were compiled and processed in a multivariate ANOVA analysis (Table 2 and Figure 2). 

Firstly, the results presented in Table 2 showed a significant influence of all studied parameters on the variation of cell survival with a probability inferior to 10^−6^ for every parameter. Indeed, in our experimental conditions, the cell survival variance was impacted (i) at 40% by the time of Olaparib exposure, (ii) at 39% by the radiotherapy dose, (iii) at 14% by the Olaparib concentration, and at 9% by the type of cell line. These results suggest clearly that both the duration of Olaparib exposure and the dose of radiotherapy play a major role in the decrease of survival of these TNBC cell lines. Secondly, the multivariate ANOVA analysis allowed to determine the impact of each Olaparib dose with or without irradiation on cell survival (Figure 2). 

The impact of all experimental parameters (type of cell line, dose of Olaparib, Olaparib time of exposure and dose of radiotherapy) on cell survival variance was analyzed in a multivariate ANOVA analysis, performed with the SEM software (Centre Jean Perrin, Clermont-Ferrand, France). The probability of variance influence in the analysis and the percentage of variance explanation were described for each experimental parameter. All parameters explained the survival variance (probabilities inferior to 10^−6^) and the strongest parameters influencing this variation were the time of Olaparib exposure (40%) and the radiotherapy dose (38%).

For this, the impact of radiotherapy dose and the time of Olaparib exposure were studied on all the data of cell survival whatever the type of cell line, after a treatment with Olaparib alone (all concentrations), radiotherapy alone and Olaparib (all concentrations) combined with irradiation (Figure 2A). The results showed the same impact on cell survival after 24 h, 72 h and 120 h of exposure to Olaparib alone, with 95 ± 1%, 80 ± 1% and 62 ± 2% or after radiotherapy alone, with 2 Gy at 100 ± 2%, 6 Gy at 81 ± 3% and 10 Gy at 62 ± 3%, respectively. In contrast, the combination of Olaparib and radiotherapy caused a significant cell survival decrease from 6 Gy and 72 h exposure with 64 ± 1%, which kept decreasing after 10 Gy and 120 h exposure with 37 ± 1%. These results showed clearly that Olaparib combined with radiotherapy induced a significant decrease in cell survival compared to radiotherapy alone, or Olaparib alone, and this, independently of the Olaparib concentration or the type of cell line (curve effect and class effects inferior to 10^−6^). 

Then, the impact of several Olaparib tested concentration was represented in Figure 2B. This analysis showed a strong decrease in survival for cells treated with 1 to 5 µM Olaparib for 120 h and 10 Gy compared to 120 h Olaparib treatment alone. Indeed, cell survival was of 48 ± 2% vs. 94 ± 3% for 1 µM and of 39 ± 2% vs. 71 ± 2% for 5 µM, respectively. More, the curve effect probabilities for each dose were systematically inferior to 10^-6^. In contrast, with higher Olaparib concentrations of 50 and 100 µM, cell survival remained relatively similar at 120 h between a treatment with Olaparib alone or combined with 10 Gy radiotherapy, with 36 ± 2 vs. 30 ± 2% for 50 µM and 31 ± 2% vs. 27 ± 2 for 100 µM, with curve effect probabilities of 0.02 and 0.0001, respectively. 

These results showed that the co-treatment Olaparib and radiotherapy was more efficient than a treatment with Olaparib alone, or radiotherapy alone. Thus, this analysis showed that the beneficial of the co-treatment (Olaparib + RX) compared to Olaparib treatment alone was more important with low Olaparib concentrations. 

### 3.3. Impact on TNBC Cell Clonogenic Potential of Repopulation 

The clonogenic survival of cells treated with Olaparib alone (concentrations of 0.5, 5 or 50 µM for 24, 72 and 120 h) or concomitantly with radiotherapy (2, 6 and 10 Gy, respectively) was determined for both MDA-MB-231 (Figure 3A) and SUM1315 (Figure 3B) cell lines cultured in monolayer. 

For MDA-MB-231 cell line, the 0.1% DMSO controls after 24 h, 72 h and 120 h incubation presented same clonogenic survivals as the controls with 89 ± 19%, 90 ± 6% and 92 ± 40%, respectively (*p* > 0.05 for all conditions) (Figure 3Aa,b). For MDA-MB-231 cell line treated with Olaparib 0.5 to 50 µM, the clonogenic survival decreased firstly during 72 h (22 ± 2% for 0.5 µM and 0.5 ± 0.1% for 50 µM, *p*= 10^−6^ et *p* = 10^−6^, respectively) and then increased again after 120 h treatment with 100 ± 3% and 1 ± 0.1% (*p* = 0.05 and *p* = 0.007, respectively) (Figure 3Aa). These results suggest that after a longtime exposure to Olaparib (120 h), the clonogenic capacity of MDA-MB-231 cell line may be restarted. Then, the radiotherapy alone induced strong clonogenic decrease with 10 ± 2% after 6 Gy to 3 ± 1% after 10 Gy, respectively (*p* = 0.0006) (Figure 3Ab). These results showed that fractioned irradiation induced a clear potential clonogenic decrease very effective at 10 Gy. More, after a co-treatment Olaparib/irradiation, this effect was amplified by the time (for 0.5 µM Olaparib, with 6 ± 2% after 6 Gy to 1 ± 0.8% after 10 Gy, *p* = 0.01) and according to Olaparib concentration with 1 ± 0.8% after 0.5 µM and 6 Gy and 0.2 ± 0.01% after 50 µM and 6 Gy (*p* = 0.03 and *p* = 0.003 compared to Rx alone, respectively) (Figure 3Ab). In contrast, the clonogenic repopulation remained the lowest and similar between 5 and 50 µM Olaparib treatment combined with 10 Gy (0.3 ± 0.2% and 0.2 ± 0.01%, respectively, *p* = 0.09). 

For SUM1315 cell line, the 0.1% DMSO controls after 24 h, 72 h and 120 h incubation presented same clonogenic survivals as the controls with 98 ± 8%, 94 ± 2% and 100 ± 18%, respectively (*p* > 0.05 for all conditions) (Figure 3Ba). For this cell line after Olaparib treatment alone, the clonogenic repopulation capacity decreased in a time and dose-dependent manner, with 65 ± 1% for 0.5 µM to 1 ± 0.1% for 50 µM after 120h (*p* = 10^−5^) (Figure 3Ba). Similarly, after irradiation, the clonogenic potential decreased strongly with 8 ± 2% after 6 Gy and 5 ± 1% after 10 Gy, compared to 2 Gy with 68 ± 14% (*p* = 0.03 and *p* = 0.03, respectively) (Figure 3Bb). More, after a co-treatment Olaparib/irradiation, this effect was intensified with 0.5 µM Olaparib and 10 Gy (0.1 ± 0.01%). Again, no difference in clonogenic survival of 5 and 50 µM treated cells was detected after 10 Gy (*p* = 0.75) (Figure 3Bb).

For both cell lines, the co-treatment with low dose of Olaparib (0.5 to 5 µM) and fractioned irradiation (10 Gy) induced very low clonogenic potential inferior to 1% indicating a nearly eradicated repopulation potential of these cell lines. 

#### Impact on Tumor-Like Models 

Thanks to preliminary 2D experiments, the long-term cell toxicity of Olaparib was highlighted as well as the interest of using low Olaparib doses (0.5–5 µM) compared to high doses (50–100 µM). Thus, for the tumor-like experiments, the spheroids were treated at long-term (6–10 days) with Olaparib alone (5 µM—low dose and 50 µM—high dose) and without/with fractioned irradiation (2, 6 and 10 Gy). For this, (i) the spheroid growth inhibition, (ii) the spheroid viability/mortality profiles and (iii) the spheroid metabolic activity were analyzed (Figure 4 and Figure 5). 

Firstly, for all treatment conditions, no difference in spheroid size and metabolic activity was detected between 0.1% DMSO-treated spheroid controls and control spheroids (Appendix A). Secondly, the impact of all treatment strategies (Olaparib alone, radiotherapy or the co-treatment) was firstly analyzed according to the treatment duration (6, 8 or 10 days treatment), and then according to Olaparib concentrations (5 or 50 µM).

1. Impact of treatment duration

For MDA-MB-231 spheroids treated with 5 µM Olaparib alone, a significant decrease of spheroid size (82 ± 16% vs. 66 ± 7%, *p* = 10^−5^) (Figure 4A–C) and metabolic activity (76 ± 10% vs. 59 ± 12%, *p* = 0.04) (Figure 4D–F) was detected from 6 to 10 days treatment duration, respectively. Similar results were highlighted after 50 µM Olaparib treatment alone (68 ± 4% vs. 35 ± 6% for spheroid size and 64 ± 2 vs. 19 ± 2% for metabolic activity, after 6 and 10 days, respectively, *p* = 10^−30^ and *p* = 10^−12^) (Figure 4A–F). In addition, after irradiation alone, spheroid size and metabolic activity decreased significantly from 108 ± 14% to 59 ± 3% (*p* = 10^−18^) and 102 ± 10% to 43 ± 6% (*p* = 10^−10^) after 6 and 10 days, respectively (Figure 4D–I). Same results were also obtained for 5 and 50 µM co-treated spheroids (with 50 µM, spheroid size of 62 ± 4% vs. 31 ± 5%, *p* = 10^−33^ and metabolic activity of 59 ± 4% vs. 21 ± 1%, *p* = 10^−7^, after 6 and 10 days of co-treatment, respectively)(Figure 4D–I). 

For SUM1315 spheroids, the comparison of spheroid size and metabolic activity, whatever the Olaparib concentration (5 or 50 µM) with or without irradiation, showed a similar significant decrease by the time (Figure 5). For instance, after 6 and 10 days of treatment with 5 µM Olaparib alone, spheroid size and metabolic activity decreased from 94 ± 9% to 71 ± 10% (*p* = 10^−12^) and from 84 ± 10% to 55 ± 2% (*p* = 0.0008), respectively (Figure 5A–F). Similarly, with 5 µM Olaparib and irradiation, the same parameters decreased significantly from 87 ± 8% to 64 ± 14% (*p* = 10^–10^) and from 100 ± 12% to 46 ± 1% (*p* = 10^−5^), respectively (Figure 5D–I). 

In addition, for both cell lines, the time-dependant decrease in terms of spheroid growth and metabolic activity was supported by an increase in the number of dead cells (red/yellow markings) within the spheroids, whatever the treatment condition (Figure 4A–C,G–I and Figure 5A–C,G–I). All these results demonstrated for both models, the importance of treating the spheroids at long-term, with Olaparib alone or combined with radiotherapy, in order to increase the tumor growth inhibition and the cell mortality. 

2. Impact of Olaparib concentrations

The same spheroid parameters were compared for both models, according to the high/low Olaparib concentrations and without/with irradiation. For MDA-MD-231 model after 10 days, a significant decrease in spheroid size and metabolic activity was detected, from 66 ± 7% (5 µM Olaparib) to 39 ± 4% (5 µM and 10 Gy, *p* = 10^−23^) and from 59 ± 12% (5 µM Olaparib) to 20 ± 1% (5 µM and 10 Gy, *p* = 0.01), respectively (Figure 4C,F,I). Similarly, SUM1315 spheroids size evolved from 71 ± 10% to 64 ± 14% (*p* = 0.04) and metabolic activity from 55 ± 2% to 46 ± 1% (*p* = 10^−7^) after the same treatment condition, respectively (Figure 5C,F,I). More, for both models, the increase in Olaparib concentration (50 µM) and 10 Gy irradiation led to low additional toxicity on spheroid size (31 ± 5%, *p* = 10^−10^ for MDA-MB-231 and 63 ± 10% for SUM1315, *p* = 0.77) and even induced an increase in spheroid metabolic activity (21 ± 1% for MDA-MB-231 and 63 ± 4%, for SUM1315, both superior to low dose Olaparib 5 µM, *p* = 0.01 and *p* = 10^−5^)(Figure 4C,F,I and Figure 5C,F,I). 

Finally, the sensitivity of each spheroid model was analyzed by the time after irradiation and 5 or 50 µM Olaparib treatment by the ANOVA analysis (Figure 6). It confirmed the efficacy of a long-term and low dose Olaparib treatment (5 µM) compared to a high dose (50 µM) (20 ± 1% and 46 ± 1% of metabolic activity for MDA-MB-231 and SUM1315 spheroids after 5 µM and 10 Gy compared to 22 ± 1% and 63 ± 3%, respectively, class effect <10^−6^). Additionnaly, whatever the treatment condition, MDA-MB-231 spheroids seemed to be more sensitive to the treatment than SUM1315 spheroids (Figure 6). All these results with tumor-like models demonstrated the potentiation of irradiation combined with Olaparib treatment and confirmed the interest of administrating a long-term co-treatment with low-dose of Olaparib. 

### 3.4. The Difference in SUM1315 and MDA-MB-231 TNBC Cell Lines Sensitivity in 2D and 3D Is Explained by a Strong Genomic Heterogeneity 

In our experiments, the sensitivity was heterogeneous between the two cell lines as depicted in the ANOVA analysis. In both 2D and 3D cell cultures, MDA-MB-231 cell line was more sensitive to the co-treatment 5 µM Olaparib/irradiation than SUM1315 cell line. Thereby, in order to explain this difference, a in silico transcriptomic comparison was carried out from GEOdatasets (Figure 6). The transcriptome of both cell lines was analyzed using a classical bioinformatics pipeline including TOPHAT and DESeq2 package. First, this analysis underlined 3870 significant differentially expressed genes (DEGs) between the two cell lines (Figure 7A). Among these DEGs, 1704 (44%) were overexpressed in MDA-MB-231 cell line whereas 2166 (56%) were overexpressed in SUM1315 cell line. 

Then, we focused on those DEGs involved in the DNA reparation pathways that play a major role in Olaparib and radiotherapy efficacy [21] (Figure 7B). Among all key selected genes, CHEK1 (log fold change LFG of 2.3, *p* < 0.01), AURKA (LFG = 2.1, *p* < 0.01), LIG1 (LFG = 1.7, *p* < 0.01), RAD51 (LFG = 1.5, *p* < 0.01) and EXO1 (LFG = 2.2, *p* < 0.01) were significantly overexpressed in MDA-MB-231 against SUM1315. Conversely, DNAPK (LFG = 1.88, *p* < 0.01), XLF (LFG = 2.82, *p* < 0.01), MRN (LFG = 5.5, *p* < 0.01), PMS2 (LFG = 2.1, *p* < 0.01) and PARP3 (LFG = 1.3, *p* < 0.01) were overexpressed in SUM1315 cell line compared to MDA-MB-231 cell line (Figure 7B). In contrast, concerning the principal targets of the PARP inhibitor Olaparib that are PARP1, and PARP2, no significant difference in gene expression was detected (Figure 7B). 

Then, for a further insight into the differential phenotypes between the two cell lines, we next used the powerful Ingenuity Pathway Analysis (IPA) software to infer the functional phenotype of each cell line from their differential transcriptomic patterns (Figure 7C). First, for both cell lines, the cell survival pathway was highly activated (z-score > 7). In contrast, an important number of cellular functions were differentially activated between these two cell lines. Indeed, for MDA-MB-231 cell line, the proliferation, migration and invasion capacities were highly activated (z-score > 3 for cellular functions), as already reported in previous studies [13]. Conversely, SUM1315 cell line did not exhibit the same cellular pattern of proliferation and migration as MDA-MB-231 but presented a particular phenotype highly associated with neuron development and morphogenesis characteristics (z-score > 2.5). These results demonstrated the stem-cell like phenotype of SUM1315 cell line, suggesting a specific neuronal engagement profile. 

## 4. Discussion

The Triple-Negative breast cancers (TNBC) are very aggressive and have a poor prognostic. TNBC cancer cells exhibit several DNA repair pathway deficiencies, such as deleterious mutations on the BRCA1/2 genes or assimilated “BRCAness” phenotypes. Thus, for this tumour subtype, the use of PARP inhibitors (PARPi), exploiting the concept of synthetic lethality, seem to be very promising [22,23,24,25,26,27]. 

In addition, the inhibition of PARPi has been described to increase the radio-sensitivity of tumour cells exhibiting a BRCAness phenotype [9,28,29,30,31]. Thus, PARPi could be administered in combination with chemotherapy targeting DNA or radiotherapy [10,29]. Indeed, PARPi induce an increase in the rate of unrepaired DNA single-strand breaks in proliferating cells, leading to the more frequent appearance of lethal DNA double-stranded breaks. Similarly, the cytotoxicity of irradiation is mainly related to the rate of DNA double-strand breaks, due to the accumulation of lethal lesions and sub lethal lesions [10]. In this context, our study aimed to model the effectiveness of a co-treatment anti-PARP (Olaparib) and fractioned irradiation, using two TNBC cell line models with metastatic origin i.e., MDA-MB-231 (highly proliferative, BRCA1-wild with BRCAness profile) and SUM1315 (low proliferative with stem-cell like phenotype, BRCA1-mutated), in 2D and 3D cell culture conditions. 

Firstly, the DNA double-strand breaks induction further to the co-treatment Olaparib 4 h and 2 Gy irradiation was determined on both models by the analysis of focigammaH2AX increase. Indeed, these proteins are the first recruited in the process of DSBs reparation by the HR system and are usually used to determine the cell sensitivity to DNA damaging agents or radiations [28,32,33]. For both studied models treated with 5 and 50 µM Olaparib alone, the number of DNA double strand breaks per nuclei increased in an Olaparib-dose-dependent-manner with 2.1- to 2.3-fold higher levels than 0.1% DMSO controls, respectively. Similarly, the number of foci per nuclei was increased by 2.4 to 3.5-fold in the presence of 5 and 50 µM Olaparib and radiotherapy, as compared to the irradiated controls. These results suggest a cumulative induction of DNA double-strand breaks further to the co-treatment, probably related to (i) the irradiation and (ii) the accumulation of unrepaired single-strand breaks by the inhibition of the BER repair system, thus degenerating into DSBs. 

Secondly, the cellular response further to the co-treatment Olaparib/fractioned irradiation was studied on both cell models cultured in 2D by cell survival and cell clonogenic repopulation tests. These experiments highlighted the low-dose efficacy of Olaparib combined with high dose of irradiation, in terms of (i) cell survival decrease and (ii) cell clonogenic repopulation capacity, compared to high dose Olaparib alone or high dose of irradiation alone. Indeed, the use of low-dose Olaparib potentialized the irradiation in terms of clonogenic repopulation decrease with a ratio “Olaparib/Olaparib + 10 Gy” of 100- to 650-fold after 0.5 µM, of 0.8- to 188-fold after 5 µM, compared to only 0.4- to 4-fold after 50 µM Olaparib (for SUM1315 and MDA-MB-231 cell lines, respectively). In the same way, other in vitro studies also showed a strong cytotoxic effect of the treatment combining hypofractioned irradiation (0 to 6 Gy) and 10 µM of PARPi Olaparib, on cholangiocarcinoma, a very malignant and radioresistant tumor model [34]. 

Our experiments in 2D cell culture conditions allowed us to choose the Olaparib concentrations (5 and 50 µM) for the treatment on more-predictive tumor-like models. More, the spheroid models allow long-term cultures up to 14 days [13]. Thus, a longer exposure time (up to 10 days) was designated in order to mimic the potential treatment strategy in vivo. For the proliferative MDA-MB-231 spheroid model, a cytostatic and cytotoxic effect of the co-treatment was detected, with a higher efficacy of the low-dose Olaparib compared to the high dose (50 µM). Indeed, the ratio of metabolic activity decrease “Ola/Ola + 10 Gy” was of 3.3-fold for 5 µM compared to 0.9 with 50 µM. Similarly, for the non-proliferative SUM1315 spheroid model, the co-treatment showed a cytotoxic activity, with always a higher efficacy of low-dose Olaparib (5 µM). The spheroid metabolic activity decreased was of 1.2-fold with 5 µM “Ola/Ola + 10 Gy” against 0.9-fold with 50 µM. These results suggest that the potentiation of the co-treatment could be reached with low-dose Olaparib and long-term exposure. More, the interest of using these tumor-like models resides also in their long-term preservation, mimicking the treatment strategies in vivo. 

In the same way, the combination PARPi/radiotherapy has been studied for the treatment of various types of cancers in preclinical studies and has demonstrated improved efficacy compared to PARPi treatments alone or radiotherapy alone. Indeed, in vivo studies evaluated the impact of Olaparib combined with fractioned radiotherapy sessions (5 x 2 Gy) on bronchial cancer xenografts, leading to delayed tumor growths [30]. An in vivo study on prostate cancer showed significant growth delay and clonogenic kill after hypofractioned or fractioned radiotherapy coupled to Olaparib [28]. Other studies showed similar results on breast cancer xenografts (MCF-7 cell line), where veliparib (another PARP inhibitor) was associated with 6 Gy of hypofractioned X-ray session [35].

Furthermore, the cell ANOVA survival analysis in both 2D and 3D cell culture conditions highlighted the heterogeneous sensitivity between the two cell line models. MDA-MB-231 cell line was more sensitive to the co-treatment than SUM1315. Thereby, a in silico transcriptomic comparison was carried out. These analyses highlighted in MDA-MB-231 the overexpression of AURKA, a gene exhibiting cell cycle regulation function and known as a biomarker of PARPi sensitivity [36,37,38,39]. Therefore, this specific overexpression of AURKA could partially explain the increased sensitivity to the co-treatment of MDA-MB-231 model compared to SUM1315 model. Thus, these results suggest that AURKA overexpression might be a potential biomarker of the co-treatment efficacy. Secondly, according to the differential transcriptomic patterns between both models, MDA-MB-231 cell line exhibited strong proliferation, migration and invasion functions, already reported in previous studies [13]. In contrast, SUM1315 cell line expressed a stem-cell like phenotype, with a specific neuronal engagement profile. This general pattern could partially explain the resistant property of SUM1315 cell line, as undifferentiated stem cells exhibit slowed proliferation rates and develop specific resistance mechanisms [40]. Otherwise, the neuronal differentiation pattern in SUM1315 cell line might be explained by its metastatic tumoral origin [41]. Indeed, Triple-Negative breast tumors frequently develop metastases in brain, compared to other types that invade preferentially bones or lungs [4]. 

## 5. Conclusions

All these preclinical modeling experiments using the more-predictive and high throughput tumor-like models suggest the perspective of a low dose and long-term Olaparib administration alongside fractioned irradiation for resistant metastatic breast cancers. Other studies on Triple-Negative Breast Cancer cases have to be performed to assess whether AURKA overexpression/amplification could predict the “PARPi and irradiation” sensitivity regardless of the proliferation/stem-cell like status and the BRCA-status. These hypothesis are also to be corroborated on other tumor-like spheroid models that are PARPi candidates. 

## Figures and Tables

**Figure 1 jcm-09-00064-f001:**
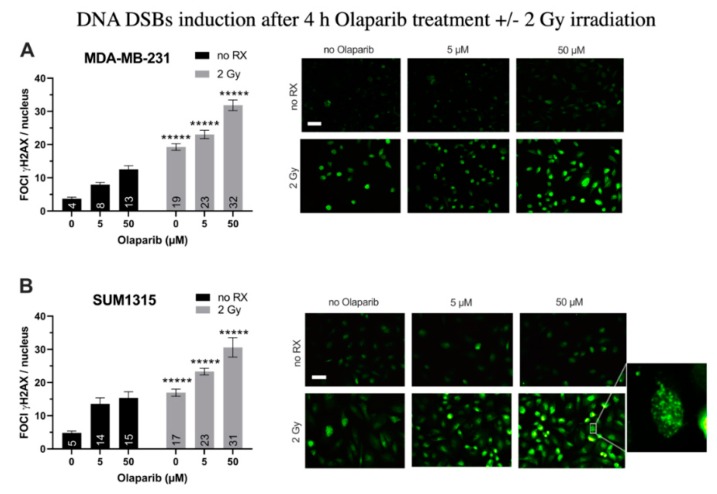
DNA Double Strand Breaks (DSBs) induction in MDA-MB-231 (**A**) and SUM1315 (**B**) cell lines cultured in monolayer after a treatment with 0.1% DMSO (control cells), 5 and 50 µM Olaparib for 4 h alone (no irradiation “no RX” ) or combined with 2 Gy of irradiation (4 h and 2 Gy). The number of foci gammaH2AX per cell in >10 fields per condition was determined with ImageJ software. Results are represented as mean number of foci gammaH2AX per nuclei with their standard deviation. A student *t*-test was performed for statistical comparison of all treatment conditions, where ns = non-significant and ***** *p* < 0.00001. Magnification = 40×, scale bar = 50 µm.

**Figure 2 jcm-09-00064-f002:**
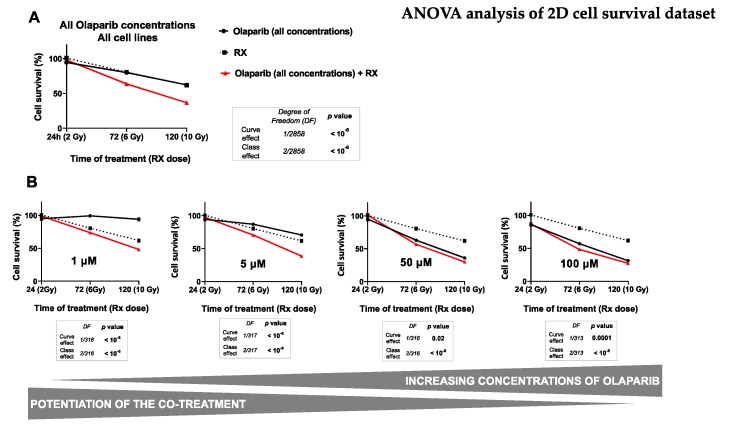
ANOVA analysis of monolayer cell culture sensitivity (SRB survival test) after the co-treatment: The graph (**A**) describes the decrease in the cell survival of both cell lines (MDA-MB-231 and SUM1315) by the time, according to (i) the Olaparib treatment alone (all concentrations), (ii) the treatment with fractioned irradiation alone or (iii) the co-treatment with Olaparib (all concentrations) and fractioned irradiation. The second set of graphs (**B**) describes the decrease of cell survival of both cell lines by the time according to (i) the Olaparib treatment alone after each concentration of 1 µM, 5 µM, 50 µM or 100 µM and (ii) the co-treatment Olaparib (same concentrations) with fractioned irradiation. For **A** and **B**, the curve effect (= significant decrease in cell survival by the time for a same condition) and the class effect (= significant difference in cell survival for a same incubation time between each condition) were determined by the ANOVA analysis, associated with their degree of freedom (= number of data of each analysis).

**Figure 3 jcm-09-00064-f003:**
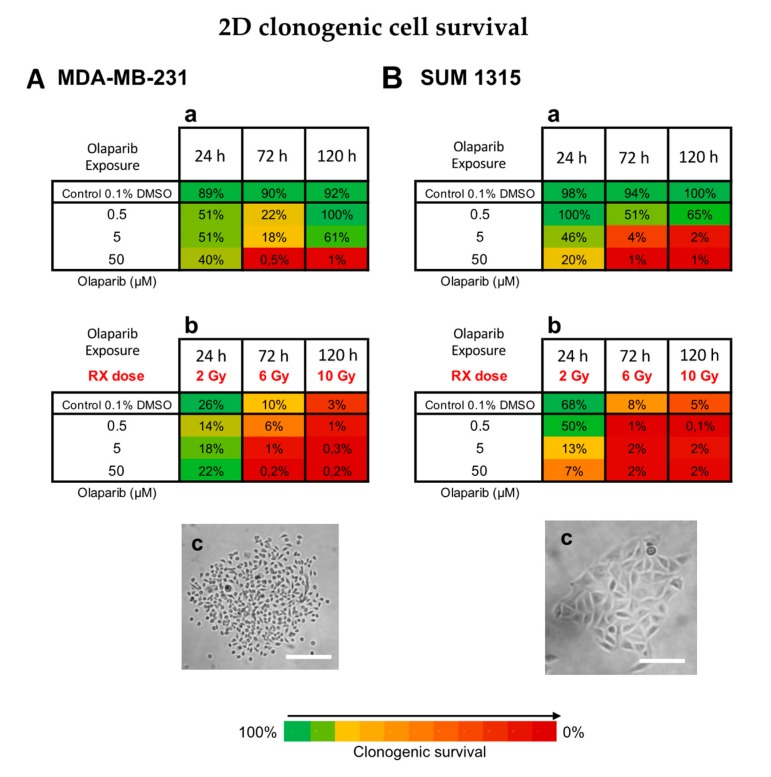
Clonogenic cell survival of MDA-MB-231 (**A**) and SUM1315 (**B**) cell lines cultured in monolayer, after a treatment with 0.5, 5 and 50 µM Olaparib alone for 24 h, 72 h, or 120 h (**a**), or combined with fractioned irradiation of 2, 6 or 10 Gy (“Rx dose”) (**b**). For each treatment condition, the percentage of the clonogenic survival was determined by the ratio of the survival fraction of treated cells on the survival fraction of 0.1% DMSO control cells. Results are represented on a Heatmap. Standard errors are <10% for every treatment condition (*n* > 3). (**c**) Photographs of clonogenic populations of MDA-MB-231 and SUM1315, Magnification = 40×, scale bar = 50 µm.

**Figure 4 jcm-09-00064-f004:**
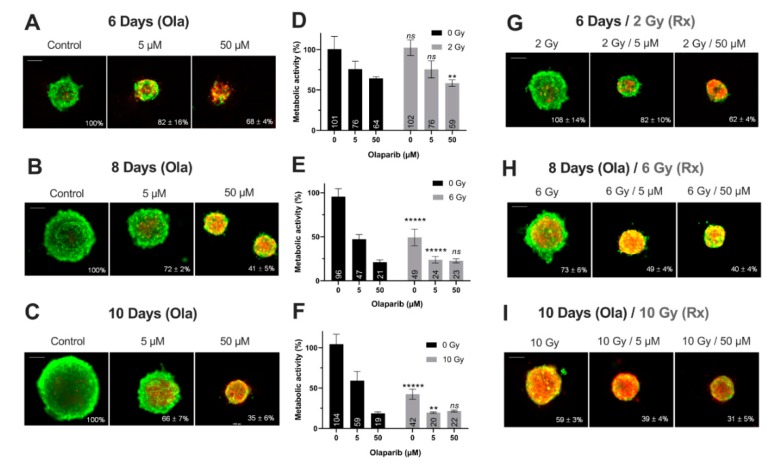
MDA-MB-231 spheroids live/dead profile (**A**–**C,G**–**I**) and metabolic activity (**D**–**F**) after a treatment with Olaparib (“Ola”, 5 and 5 µM) for 6, 8 and 10 days with or without fractioned irradiation (“Rx”). (**A**–**C,G**–**I**) Spheroid viability/mortality profile and spheroid normalized size after a treatment with Olaparib and/or fractioned irradiation for 6 days (**A**,**G**), 8 days (**B**,**H**) and 10 days (**C**,**I**): Images were taken with Cytation3MV where green markings correspond to calcein-AM penetration (viable cells) and red markings correspond to ethidium homodimer-1 cell penetration (dead cells). The percentage of spheroid size was normalized by the size of spheroid controls. Scale bar = 200 µm. (**D**–**F**) MDA-MB-231 cell metabolic activity in spheroids with resazurin test after 6 (**D**), 8 (**E**) and 10 days of treatment (**F**). Corrected fluorescence intensity of resorufin (ex/em λ576/λ584 nm) was measured after 15 h incubation with 60 µM resazurin. The fluorescence intensity of treated spheroids (*n* > 15) was normalized by the fluorescence intensity of each associated 0.1% DMSO control spheroids. Results are expressed as mean metabolic activity (%) of each treatment conditions with their standard errors. A student *t*-test was performed for the statistical comparison of all treatment conditions and the *p*-value is represented in the graphs as ns = non-significant, ** *p* < 0.01 and ***** *p* < 0.00001.

**Figure 5 jcm-09-00064-f005:**
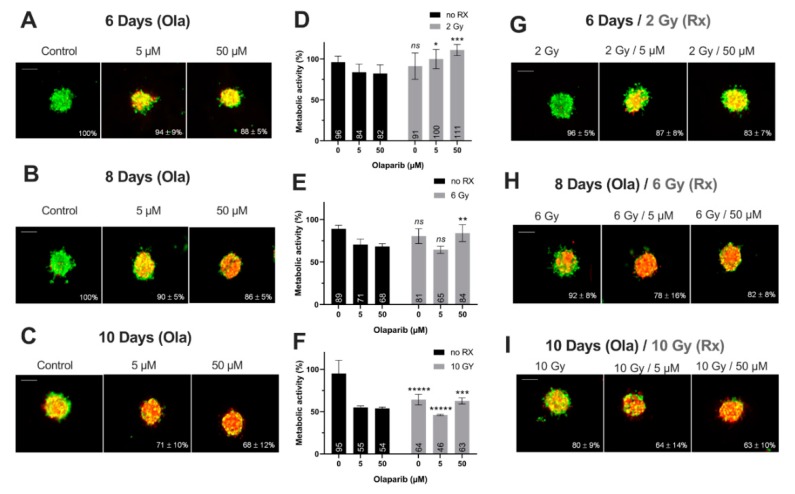
SUM1315 spheroids live/dead profile (**A**–**C,G**–**I**) and metabolic activity (**D**–**F**) after a treatment with Olaparib (“Ola”, 5 and 5 µM) for 6, 8 and 10 days with or without fractioned irradiation (“Rx”). (**A**–**C,G**–**I**) Spheroid viability/mortality profile and spheroid normalized size after a treatment with Olaparib and/or fractioned irradiation for 6 days (**A**,**G**), 8 days (**B**,**H**) and 10 days (**C**,**I**): Images were taken with Cytation3MV where green markings correspond to calcein-AM penetration (viable cells) and red markings correspond to ethidium homodimer-1 cell penetration (dead cells). The percentage of spheroid size was normalized by the size of spheroid controls. Scale bar = 200 µm. (**D**–**F**) SUM1315 cell metabolic activity in spheroids with resazurin test after 6 (**D**), 8 (**E**) and 10 days of treatment (**F**). Corrected fluorescence intensity of resorufin (ex/em λ576/λ584 nm) was measured after 15 h incubation with 60 µM resazurin. The fluorescence intensity of treated spheroids (*n* > 15) was normalized by the fluorescence intensity of each associated 0.1% DMSO control spheroids. Results are expressed as mean metabolic activity (%) of each treatment conditions with their standard errors. A student t-test was performed for the statistical comparison of all treatment conditions and the p value is represented in the graphs as ns = non-significant, * *p* < 0.05, ** *p* < 0.01, *** *p* < 0.001 and ***** *p* < 0.00001.

**Figure 6 jcm-09-00064-f006:**
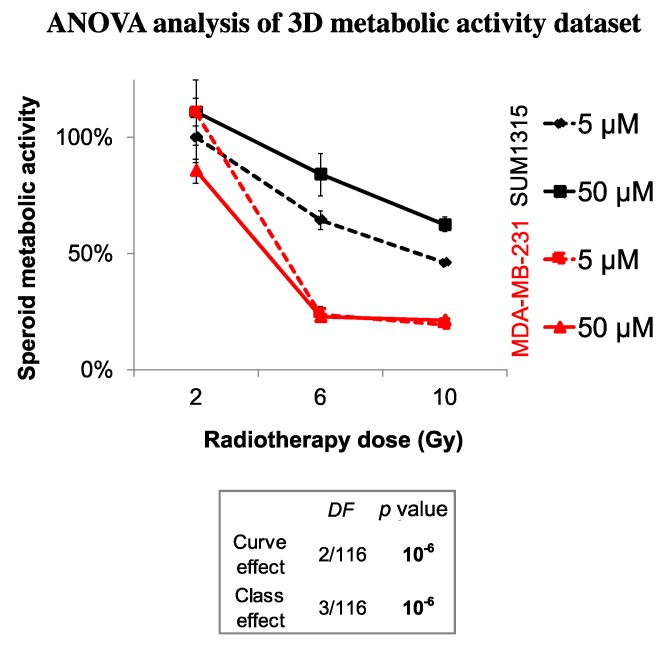
ANOVA analysis of spheroids sensitivity to the co-treatment with Olaparib and 2 to 10 Gy of fractioned irradiation: The decrease in spheroid metabolic activity is potentialized with 5 µM Olaparib and heterogenous between the two MDA-MB-231 and SUM1315 cell lines. The curve effect indicates a significant decrease in cell survival by the time for a same condition and the class effect indicates a significant difference in cell survival for a same incubation (DF = Degree of Freedom).

**Figure 7 jcm-09-00064-f007:**
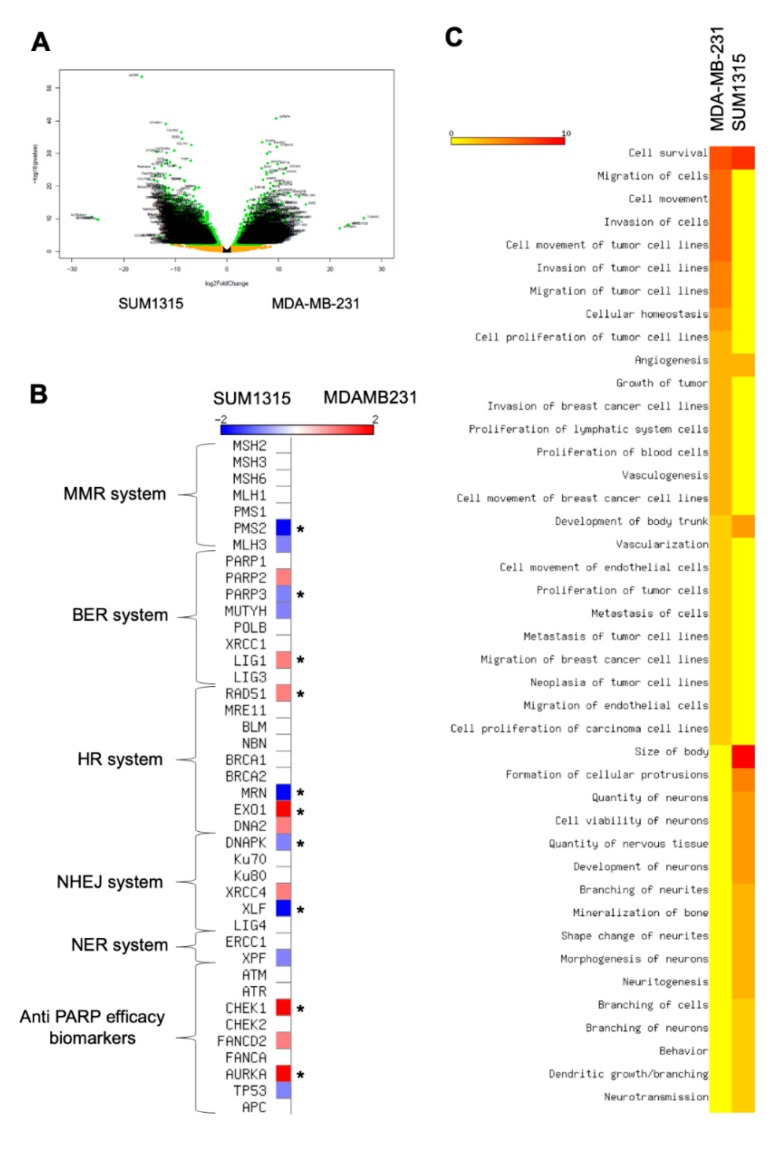
Transcriptomic analysis of MDA-MB-231 and SUM1315 cell lines: Determination of cell difference in sensitivity by a Differential Expressed Gene analysis (**A**,**B**) and a functional phenotype analysis (**C**). (**A**) Volcanoplot of the Differential Expressed Genes between MDA-MB-231 and SUM1315 cell lines. The y-axis corresponds to the mean expression value of log 10 (*p*-value), and the x-axis displays the log 2 fold change value. Each point represents a gene. The green dots represent the significant up/down expressed genes (adjusted *p* value < 0.05, log 10 >1) whereas the orange dots represent the non-significant differentially significant genes. Genes symbol are indicated. (**B**) Expression levels of keys genes involved in DNA reparation mechanisms (mismatch repair system “MMR”, base excision repair system “BER”, homologous recombination system “HR”, non homologous end-joining repair system “NHEJ”, nucleotide excision repair system “NER”). The heatmap shows the expression level of key genes involved in different DNA reparation pathways. These last are indicated beside the heatmap. Asterisk marks designated genes which are significantly differential expressed genes between the two cell lines: MDA-MB-231 and SUM1315, * *p* < 0.01. (**C**) Determination of MDA-MB-231 and SUM1315 cell lines functional phenotype by an Ingenuity Pathway (IPA) analysis. The heatmap shows the cellular functions that were inferred to be associated with enriched DGE clusters between MDA-MB-231 and SUM1315 cell lines.

**Table 1 jcm-09-00064-t001:** Accession number, sample and sequencing information from the NCBI public GEO DataSet.

Name	Accession N	Library Type	Sequencing	End Type	Reference
MDAMB231_A	GSE73526	TruSeq Stranded mRNA	Illumina HiSeq 2000	paired-end	[17]
MDAMB231_B	GSE48213	TruSeq RNA Sample Preparation Kit	Illumina Genome Analyzer IIx	paired-end	[18]
MDAMB231_C	GSE83132	TruSeq RNA Sample Prep Kit v2	Illumina HiSeq 2000	paired-end	[19]
MDAMB231_D	GSE83132	TruSeq RNA Sample Prep Kit v2	Illumina HiSeq 2000	paired-end	[19]
MDAMB231_E	GSE83132	TruSeq RNA Sample Prep Kit v2	Illumina HiSeq 2000	paired-end	[19]
SUM1315_B	GSE48213	TruSeq RNA Sample Preparation Kit	Illumina Genome Analyzer IIx	paired-end	[18]
SUM1315_A	GSE73526	TruSeq Stranded mRNA	Illumina HiSeq 2000	Illumina HiSeq 2000	[17]

**Table 2 jcm-09-00064-t002:** Multivariate ANOVA results obtained from the SRB monolayer cell survival dataset.

Experimental Parameters	Probability of Variance Influence	% of Variance Explanation
Type of cell line (MDA-MB-231; SUM1315)	<0.0000001	8%
Dose Olaparib (0; 0.1; 1; 5; 10; 25; 50; 75; 100 µM)	<0.0000001	14%
Olaparib exposure (24 h; 72 h; 120 h)	<0.0000001	40%
Irradiation dose (0; 2; 6; 10 Gy)	<0.0000001	38%

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
