# Peer review of "Low-Dose and Long-Term Olaparib Treatment Sensitizes MDA-MB-231 and SUM1315 Triple-Negative Breast Cancers Spheroids to Fractioned Radiotherapy"

_jcm, 2019, doi:10.3390/jcm9010064_

Round 1
Reviewer 1 Report
Regarding this manuscript, I feel it is a well-written manuscript with good and clear data to convince readers about their findings. The abstract could be more precise to make their points more clear. I don't think the statistic analysis for the significant difference needs so many asterisks. Scale bars need to be provided in the microscopic images.
Author Response
Dear Lyanna Li,
Thanks for the reviewing.
Please find attached the responses to the reviewers concerning the manuscript jcm-668742 entitled “Low-dose and long-term Olaparib treatment sensitizes MDA-MB-231 and SUM1315 Triple-Negative Breast Cancers spheroids to fractioned radiotherapy.”
Authors: Clémence Dubois, Fanny Martin, Chervin Hassel, Florian Magnier, Pierre Daumar, Corinne Aubel, Sylvie Guerder, Emmanuelle Mounetou, Frédérique Penault-Lorca, Mahchid Bamdad *
We proceeded to a step-by-step response for each reviewer comment, that you can find below. In the revised manuscript version that we uploaded, our changes are indicated for each reviewer (Reviewer 1 / R1 or reviewer 2 / R2) using the “Track Changing” function of Microsoft Word.
Hoping this revision version will meet your expectations,
Kind regards,
Mahchid Bamdad
Mahchid.bamdad@uca.fr
Response to reviewer 1
Open Review
(x) I would not like to sign my review report
( ) I would like to sign my review report
English language and style
( ) Extensive editing of English language and style required
( ) Moderate English changes required
( ) English language and style are fine/minor spell check required
(x) I don't feel qualified to judge about the English language and style
|
Yes |
Can be improved |
Must be improved |
Not applicable |
|
|
Does the introduction provide sufficient background and include all relevant references? |
(x) |
( ) |
( ) |
( ) |
|
Is the research design appropriate? |
(x) |
( ) |
( ) |
( ) |
|
Are the methods adequately described? |
(x) |
( ) |
( ) |
( ) |
|
Are the results clearly presented? |
(x) |
( ) |
( ) |
( ) |
|
Are the conclusions supported by the results? |
(x) |
( ) |
( ) |
( ) |
Comments and Suggestions for Authors
Regarding this manuscript, I feel it is a well-written manuscript with good and clear data to convince readers about their findings.
-> The abstract could be more precise to make their points more clear.
Authors responses -> We modified the abstract as asked.
-> I don't think the statistic analysis for the significant difference needs so many asterisks.
Authors response -> As commented in the manuscript (Page 5, line 236), our tests allow us to have such level of significance, as we have little variations between the samples in each treatment condition.
-> Scale bars need to be provided in the microscopic images.
Authors response -> We added the scale bars in Figure 1 where they were missing and indicated in the Figure 1 legend in the manuscript “scale bar = 50 µm” (page 7, line 262).

Reviewer 2 Report
The work by Dubois et al. provides a thorough assessment of the possible combination of olaparib and radiotherapy in two triple negative breast cancer cell lines with different BRCA status. The author analyze through classic and reliable experiments which are the best time and doses of co-treatment using both 2D and spheroid models. As MDAMB231 were found more sensitive to the co-treatment with respect to the SUM1315, the authors analyzed possible molecular determinants trough the analysis of gene expression in public dataset. The results from this analysis are interesting but the authors should emphasize more that further studies are necessary to assess whether these findings can be generalized to other triple negative cases (ie the association of the stem cell like phenotype or AURKA expression). Beyond this I have minor observations:
-page 6 line 216 add gH2AX into the text, to specify which foci are being assessed;
-please check the use of exposure versus exposition throughout the manuscript. The first sound more appropriate to me;
-Page 22 line 462 please change to ‘and have a poor prognosis’
-Page 22 line 483 foci is written in capitals
Author Response
Dear Lyanna Li,
Thanks for the reviewing.
Please find attached the responses to the reviewers concerning the manuscript jcm-668742 entitled “Low-dose and long-term Olaparib treatment sensitizes MDA-MB-231 and SUM1315 Triple-Negative Breast Cancers spheroids to fractioned radiotherapy.”
Authors: Clémence Dubois, Fanny Martin, Chervin Hassel, Florian Magnier, Pierre Daumar, Corinne Aubel, Sylvie Guerder, Emmanuelle Mounetou, Frédérique Penault-Lorca, Mahchid Bamdad *
We proceeded to a step-by-step response for each reviewer comment, that you can find below. In the revised manuscript version that we uploaded, our changes are indicated for each reviewer (Reviewer 1 / R1 or reviewer 2 / R2) using the “Track Changing” function of Microsoft Word.
Hoping this revision version will meet your expectations,
Kind regards,
Mahchid Bamdad
Mahchid.bamdad@uca.fr
Response to reviewer 2
Open Review
(x) I would not like to sign my review report
( ) I would like to sign my review report
English language and style
( ) Extensive editing of English language and style required
( ) Moderate English changes required
(x) English language and style are fine/minor spell check required
( ) I don't feel qualified to judge about the English language and style
|
Yes |
Can be improved |
Must be improved |
Not applicable |
|
|
Does the introduction provide sufficient background and include all relevant references? |
(x) |
( ) |
( ) |
( ) |
|
Is the research design appropriate? |
(x) |
( ) |
( ) |
( ) |
|
Are the methods adequately described? |
(x) |
( ) |
( ) |
( ) |
|
Are the results clearly presented? |
( ) |
(x) |
( ) |
( ) |
|
Are the conclusions supported by the results? |
(x) |
( ) |
( ) |
( ) |
Comments and Suggestions for Authors
The work by Dubois et al. provides a thorough assessment of the possible combination of olaparib and radiotherapy in two triple negative breast cancer cell lines with different BRCA status. The author analyze through classic and reliable experiments which are the best time and doses of co-treatment using both 2D and spheroid models. As MDAMB231 were found more sensitive to the co-treatment with respect to the SUM1315, the authors analyzed possible molecular determinants trough the analysis of gene expression in public dataset.
à The results from this analysis are interesting but the authors should emphasize more that further studies are necessary to assess whether these findings can be generalized to other triple negative cases (ie the association of the stem cell like phenotype or AURKA expression).
Authors response -> We added these sentences in the manuscript:
Page 24 lines 574-575: “Thus, these results suggest that AURKA overexpression might be a potential biomarker of the co-treatment efficacy.”
Page 25 lines 590-593: “Other studies on Triple-Negative Breast Cancer cases have to be performed to assess whether AURKA overexpression/amplification could predict the “PARPi & irradiation” sensitivity regardless of the proliferation/stem-cell like status and the BRCA-status.”
Beyond this I have minor observations:
-page 6 line 216 add gH2AX into the text, to specify which foci are being assessed;
Authors response -> This was added in the revised manuscript (page 6, line 244)
-please check the use of exposure versus exposition throughout the manuscript. The first sound more appropriate to me;
Authors response -> We changed all the occurrences of “exposition” in the manuscript and in the figures by “exposure”.
-Page 22 line 462 please change to ‘and have a poor prognosis’
Authors response -> Page 23 line 504, the sentence “The Triple-Negative breast cancers (TNBC) are very aggressive and of poor prognostic” was modified to “The Triple-Negative breast cancers (TNBC) are very aggressive and have a poor prognostic”.
-Page 22 line 483 foci is written in capitals
Authors response -> All occurrences of “foci” that were in in capital letters in the manuscript were changed “foci” in lowercase letters.
